# Immunomodulatory Effect of Human Lactoferrin on Toll-like Receptors 2 Expression as Therapeutic Approach for Keratoconus

**DOI:** 10.3390/ijms232012350

**Published:** 2022-10-15

**Authors:** Uxía Regueiro, Maite López-López, Rubén Varela-Fernández, Tomás Sobrino, Elio Diez-Feijoo, Isabel Lema

**Affiliations:** 1Corneal Neurodegeneration Group (RENOIR), Clinical Neurosciences Research Laboratory (LINC), Health Research Institute of Santiago de Compostela (IDIS), 15706 Santiago de Compostela, Spain; 2Department of Surgery and Medical-Surgical Specialties, Faculty of Optics and Optometry, University of Santiago de Compostela (USC), 15705 Santiago de Compostela, Spain; 3Department of Pharmacology, Pharmacy and Pharmaceutical Technology, University of Santiago de Compostela (USC), 15705 Santiago de Compostela, Spain; 4NeuroAging Laboratory (NEURAL), Clinical Neurosciences Research Laboratory (LINC), Health Research Institute of Santiago de Compostela (IDIS), 15706 Santiago de Compostela, Spain; 5Galician Institute of Ophthalmology (INGO), Conxo Provincial Hospital, 15706 Santiago de Compostela, Spain

**Keywords:** keratoconus, lactoferrin, ocular surface disorders, Toll-like receptors

## Abstract

Keratoconus (KC) is a corneal disorder whose etiology shares a close relationship with Lactoferrin (LTF) dysregulation and Toll-like Receptors 2 (TLR2) overexpression. This study shows how these two important biomarkers are clinically and molecularly interrelated, increasing knowledge about KC pathophysiology, and opening the door to future therapies. In this prospective clinical study, serum and tear LTF concentrations were quantified in 90 KC patients and 60 controls. A correlation analysis with multiple blood and tear immunoinflammatory mediators, and KC-associated tomographic parameters, was performed. An in vitro study using HEK-Blue^TM^hTLR2 cell cultures was also conducted to determine the expression and functionality of TLR2 under the influence of LTF treatment. As a result, a LTF decreased was observed in KC patients compared to controls (*p* < 0.0001), evidencing the strong correlation with TLR2 overexpression at systemic and ocular surface level, with inflammatory mediator upregulation and with KC severity. In stimulated cell cultures, TLR2 expression was decreased using 2 mg/mL of LTF. The levels of secreted embryonic alkaline phosphatase (SEAP) and interleukin-8 (IL-8) were also reduced in supernatants after LTF treatment. As conclusions, the dysregulation of LTF and TLR2 in the ocular surface of KC patients contributes to KC severity by maintaining a detrimental chronic immune–inflammatory state. The immunomodulatory properties of LTF on TLR2 expression suggest its potential as a therapeutic approach for KC.

## 1. Introduction

Lactoferrin, also named lactotransferrin (LTF), is an iron-binding mammalian glycoprotein that belongs to the transferrin family. This multifunctional glycoprotein is present in many fluids (such as tears, saliva, serum, milk, etc.) and exocrine secretions that recover mucosal sites considered as portals of entry and/or invasion of pathogens, contributing to the primary innate-immune defensive system [1,2,3]. It has an approximated molecular weight of 80 kDa and a highly conserved three-dimensional structure with iron-binding sites. Although its ability to bind Fe3+ ions allows it to play a predominant role in regulating free iron levels in body fluids, the major functions of LTF are related to antioxidant, antibacterial, and antiviral activities [4]. It can be found as an iron-free protein (apo-LTF) or a fully iron-loaded protein (holo-LTF), showing the ability to bind to a greater variety of molecules in the apo state [5]. 

On the ocular surface, LTF is secreted mainly by the lacrimal gland, but the epithelial cells of the cornea and conjunctiva also produce detectable amounts [6]. It represents 25% of the total tear proteins [7] and provides essential protection in the ocular surface tissues due to its anti-inflammatory, immunomodulatory, and antimicrobial iron-binding role [8]. Previous studies have found variations with aging and several inflammatory ocular diseases, such as dry eye or vernal conjunctivitis, being however invariable with the sex [9]. In this line, patients with keratoconus (KC) have shown a lower concentration of LTF in tear fluid than control subjects; moreover, this tear LTF reduction in KC seems to be related to the disease development [10,11]. In advanced KC, the presence of Fleischer’s ring (iron deposits around the base of the cone, localized in the epithelial basement membrane) is a clear indicator of altered iron metabolism, which may contribute to an oxidative microenvironment and cell damage, or even death processes. However, despite recent advances in understanding the LTF impact on cellular behavior, it is still unclear which pathways are activated and how this affects the epithelial and immune cell responses [12]. LTF constitutes a promising diagnostic and therapeutic target for numerous diseases; in fact, more and more research is focusing on using LTF as an active targeting ligand for drug delivery systems [13]. 

KC is a primary corneal ectasia that course with weakening, thinning, protrusion, and chronic degeneration of the corneal tissue; leading to a significant loss of vision and a reduction in quality of life [14,15]. The multifactorial etiology of KC is associated with environmental, biomechanical, genetic, and biochemical mechanisms that share a close relationship with the primary innate immune defensive system. In this line, the overexpression of Toll-like receptor 2 (TLR2) in blood monocytes and neutrophils, as well as in corneal and conjunctival epithelial cells of KC patients, was recently demonstrated [16,17]. TLR2 is an immune innate transmembrane protein that detects the presence of exogenous and/or endogenous agents associated with cell damage [18], promoting an inflammatory response and involving the molecular drivers which may cause tissue degradation in KC. The systemic innate immune TLR2 overexpression observed in patients with KC was correlated with the increase in inflammatory mediators and NF-kB factors in serum [16]. Furthermore, a study conducted in subclinical KC patients demonstrated that corneal and conjunctival TLR2 are suitable biomarkers for early detection of the disease [17].

Based on all these previous findings, which evidenced the LTF role in KC and the importance of TLR2 in the disease development; our hypothesis supports that both mediators may be intimately associated, leading to the development and maintenance of a detrimental status in KC. Therefore, the main purpose of this study was to measure serum and tear LTF concentrations in KC patients and examine their relationship with the disease’s immune–inflammatory status, and KC severity. Furthermore, an in vitro study with a stimulated HEK-Blue^TM^ human TLR2 (hTLR2) cell culture was also conducted to determine the immunomodulatory capacity of LTF on TLR2 expression.

## 2. Results

### 2.1. Clinical Study

Blood sample analysis involved 20 control subjects (55% males; mean age, 30.5 ± 7.6 years) and 40 patients with KC (55% males; mean age, 33.1 ± 10.9 years). No sex- and age-related statistical differences were detected between these controls and KC patients (*p* = 0.609 and *p* = 0.276, respectively). Serum LTF concentrations were statistically lower in KC patients than in control subjects (446 ± 331 ng/mL vs. 1187 ± 282 ng/mL, respectively) (*p* < 0.0001) Figure 1.

Tear fluid evaluation involved 40 control subjects (46% males; mean age, 29.4 ± 6.7 years) and 50 patients with KC (66% males; mean age, 33.2 ± 9.4 years). No sex- and age-related statistical differences were detected between these controls and KC patients (*p* = 0.063 and *p* = 0.105, respectively). LTF concentrations in tear fluid were also statistically reduced in KC patients compared to controls (0.83 ± 0.34 mg/mL vs. 1.28 ± 0.64 mg/mL, respectively) (*p* < 0.0001), as shown in Figure 1. In addition, tears analysis of KC patients revealed that there were no differences in LTF levels based on the presence or absence of ocular itching (0.83 ± 0.34 mg/mL vs. 0.82 ± 0.36 mg/mL, respectively) (*p* = 0.925), and based on the presence or absence of ocular rubbing (0.84 ± 0.23 mg/mL vs. 0.82 ± 0.41 mg/mL, respectively) (*p* = 0.809).

No differences were observed between the participants with or without allergic diseases for both serum and tear LTF concentrations (721 ± 487 ng/mL vs. 665 ± 462 ng/mL (*p* = 0.648), and 0.94 ± 0.40 mg/mL vs. 1.01 ± 0.50 mg/mL (*p* = 0.305); (respectively).

The bivariate correlation analysis showed a strong negative correlation between serum LTF concentrations and TLR2 expression in monocytes and neutrophils. Moreover, a strong negative correlation was also observed between serum LTF concentrations and serum inflammatory mediators such as IL-1β, IL-6, TNF-α, and MMP-9 (all *p* < 0.0001). Figure 2 shows the Pearson’s coefficients of the bivariate correlation study, representing it by scatter plots.

Moreover, a negative correlation between tear LTF concentrations and TLR2 expression in corneal and conjunctival epithelial cells was found. In this regard, a negative correlation was also observed between tear LTF concentrations and the KC severity parameters: high I-S asymmetry, coma, coma-like, and posterior elevation values. The Pearson’s coefficients of this bivariate correlation analysis are shown in Table 1. 

### 2.2. In Vitro Study

HEK-Blue^TM^hTLR2 cell cultures, a line specifically designed for monitoring TLR2 agonists and antagonists, were used in this study as a first approach to assess the immunomodulatory effect of LTF on TLR2 expression. Figure 3A shows a 20× magnification brightfield microscopy image of HEK-Blue^TM^hTLR2 and HEK-Blue^TM^Null1 cell cultures.

TLR2 functionality after stimulation, neutralization, and immunomodulation was firstly determined by measuring SEAP production in cell culture supernatants by QUANTI-Blue^TM^ assay, as shown in Figure 3B. As result, statistical differences were found between stimulated HEK-Blue^TM^hTLR2 (with Pam2CSK4 or Pam3CSK4) and unstimulated HEK-Blue^TM^hTLR2 (*p* < 0.0001), as well as between stimulated HEK-Blue^TM^hTLR2 (with Pam2CSK4 or Pam3CSK4) and unstimulated or stimulated HEK-Blue^TM^Null1 (*p* < 0.0001); demonstrating that the stimulation with both synthetic lipopeptides TLR2 agonists (Pam2CSK4 or Pam3CSK4) causes SEAP overexpression in the supernatant of HEK-Blue^TM^hTLR2 cell cultures. The pre-incubation with Anti-hTLR2-IgA reduces SEAP concentrations (*p* < 0.001 regarding Pam2CSK4, and *p* = 0.004 regarding Pam3CSK4) in the supernatant of HEK-Blue^TM^hTLR2, proving the neutralizing role of this TLR2 antagonist. Likewise, the immunomodulatory action of 2 mg/mL of human LTF was also demonstrated; so, the pre-incubation with LTF reduces SEAP levels in the supernatant of HEK-Blue^TM^hTLR2 (*p* < 0.001 regarding Pam2CSK4, and *p* = 0.004 regarding Pam3CSK4). No statistical differences were found between the pre-incubation with Anti-hTLR2-IgA and with LTF. Appendix A includes information about HEK-Blue^TM^hTLR2 pre-incubation with several LTF concentrations for the selection of the suitable dose of LTF, Appendix A.

TLR2 protein expression after stimulation, neutralization, and immunomodulation of HEK-Blue^TM^ cell cultures was evaluated by flow cytometry, immunohistochemistry, and western blot (WB) Figure 4. Flow cytometry results, presented in Figure 4A, showed that: (1) HEK-Blue^TM^hTLR2 stimulation with both TLR2 agonists (Pam2CSK4 & Pam3CSK4) triggers the TLR2 protein overexpression (statistical differences were found between unstimulated HEK-Blue^TM^hTLR2 and Pam2CSK4 (*p* < 0.0001) or Pam3CSK4 (*p* < 0.0001) stimulated HEK-Blue^TM^hTLR2 cell cultures); (2) the pre-incubation with Anti-hTLR2-IgA or with 2 mg/mL of LTF reduces the TLR2 expression in the stimulated HEK-Blue^TM^hTLR2 cell cultures (statistical differences were found between Anti-hTLR2-IgA and Pam2CSK4 (*p* < 0.0001) or Pam3CSK4 (*p* = 0.009), and between LTF and Pam2CSK4 (*p* = 0.049) or Pam3CSK4 (*p* = 0.014)); (3) there are differences between the neutralization provided by Anti-hTLR2-IgA and the immunomodulation provided by LTF in HEK-Blue^TM^hTLR2 cell cultures (being *p* < 0.0001 for Pam2CSK4 group and *p* = 0.785 for Pam3CSK4 group). These results are in accordance with QUANTI-Blue^TM^’s assay findings.

The immunohistochemical assay carried out in HEK-Blue^TM^hTLR2 cell culture confirms the high TLR2 expression in Pam2CSK4 and Pam3CSK4 stimulated cells (shown by dark brown staining), as well as the low TLR2 expression in unstimulated cells (without brown staining), Figure 4B. Comparing the Anti-hTLR2-IgA pre-incubated cell cultures and the stimulated ones, it is possible to observe a lightening of the dark brown staining, indicating that Anti-hTLR2-IgA pre-incubation contributes to TLR2 neutralization. Likewise, pre-incubation with LTF also showed a lightening of the dark brown staining seen in the stimulated cell cultures, being significantly more noticeable for the Pam3CSK4 stimulated group. Moreover, there were fewer color differences between Anti-hTLR2-IgA and LTF in the Pam3CSK4 stimulated group compared with the Pam2CSK4 stimulated group. These findings were corroborated by the quantification of the staining intensity, which was also consistent with the flow cytometer results.

The WB study was limited to the HEK-Blue^TM^ cell cultures stimulated with Pam3CSK4 because they showed better Anti-hTLR2-IgA neutralization and LTF immunomodulation capacities, Figure 4C. As a result, once again it was observed that: (1) the stimulation with TLR2 agonist triggered TLR2 overexpression (*p* < 0.0001); (2) the pre-incubation with TLR2 antagonist contributed to TLR2 neutralization (*p* < 0.0001); and (3) the pre-incubation with LTF led to TLR2 immunomodulation (*p* < 0.0001).

Finally, TLR2 functionality was determined by measuring IL-8 concentrations in the supernatants of unstimulated, stimulated, neutralized, and immunomodulated HEK-Blue^TM^ cell cultures, Figure 5. As we expect, IL-8 concentration was increased after stimulation with Pam2CSK4 or Pam3CSK4 in HEK-Blue^TM^hTLR2 (*p* < 0.0001 and *p* < 0.0001, respectively). Pre-incubation with Anti-hTLR2-IgA or with 2 mg/mL of LTF in HEK-Blue^TM^hTLR2 reduced IL-8 concentrations compared to stimulated cells (*p* = 0.002 regarding Pam2CSK4 and *p* < 0.0001 regarding Pam3CSK4, for Anti-hTLR2-IgA; *p* = 0.009 regarding Pam2CSK4 and *p* < 0.0001 regarding Pam3CSK4, for LTF). In addition, HEK-Blue^TM^hTLR2 cells stimulated with Pam3CSK4 expressed more IL-8 than HEK-Blue^TM^hTLR2 cells stimulated with Pam2CSK4 (*p* < 0.0001). No statistical differences were found between IL-8 concentrations of pre-incubated Anti-hTLR2-IgA and LTF HEK-Blue^TM^hTLR2 cells (being *p* = 0.442 for Pam2CSK4 group, and *p* = 0.168 for Pam3CSK4 group).

## 3. Discussion

This translational study aimed to examine LTF levels in serum and tear fluid of KC patients, as well as to assess its relationship with the systemic and local immune–inflammatory environment and the clinical status observed in KC. Remarkably, serum and tear LTF concentrations were found to be lower in KC patients than in control subjects. Moreover, the lower serum LTF concentrations observed in KC patients were strongly correlated with the increase in immune–inflammatory biomarkers, such as TLR2 and cytokines, in blood samples. Likewise, the lower tear LTF concentrations in KC patients were also correlated with increased immune biomarkers at the ocular surface and with high quantitative corneal parameters associated with the disease severity. In addition, the in vitro study performed in HEK-Blue^TM^hTLR2 cell cultures confirmed the immunomodulatory role of LTF on the expression and functionality of TLR2.

LTF is an iron-binding protein of the transferrin family that plays an important defensive action due to its multifunctionality [8]. In plasma, LTF derives from neutrophils and its normal concentration is very low (0.5–2 µg/mL), especially when compared to LTF concentration in the tear fluid (1–2 mg/mL) [19]. This is evidence of the important role that LTF plays on the ocular surface, since it constitutes a high percentage of the total tear proteins [7]. In this study, we compared serum LTF concentrations in patients with KC and control subjects. As a result, we observed that serum LTF levels were 2.6 times significantly lower in KC patients than in controls. In these patients, reduced serum LTF levels were strongly correlated with the TLR2 overexpression in blood monocytes and neutrophils. Furthermore, these low LTF levels found in KC patients were also correlated with the overexpression of several serum inflammatory mediators (IL-1β, IL-6, TNF-α, MMP-9). These results show that the systemic immune–inflammatory alteration observed in patients with KC is closely related to low levels of LTF.

To check what was happening at the ocular level, in this study we also evaluated whether the LTF concentration in tears correlated with the immune alteration of the ocular surface in KC patients. In this way, we observed that tear LTF levels were 1.54 times significantly lower in KC patients than in control subjects. Moreover, the tear LTF levels in KC patients were negatively correlated with the TLR2 overexpression in corneal and conjunctival epithelial cells and with some quantitative topographic, aberrometric, and tomographic parameters related to the KC severity (I-S asymmetry, coma, coma-like, and posterior elevation). All of these findings provide evidence for the LTF role in the development and progression of KC.

Mainly due to its iron uptake capacity, showing an iron affinity 300 times higher than transferrin, LTF plays an anti-inflammatory, immunomodulatory, and antimicrobial iron-binding defensive role [8]. Iron is an important electron transfer mediator which is required for essential cellular functions such as respiration, oxygen transport, DNA synthesis, energy production, and cell proliferation [20]. In a ferrous state, iron acts as an electron donor, while in a ferric state, it acts as an acceptor. Although iron is necessary for many biological processes, excessive amounts of iron are toxic and lead to oxidative stress and synthesis of highly reactive radicals, producing tissue damage [21]. In this regard, a characteristic biomicroscopic sign present in corneas with moderate or advanced KC disease is the Fleischer ring. The Fleischer ring consists of iron deposits in the epithelial basement membrane, localized surrounding the base of the cone in keratoconic corneas. Gass et al. [22] suggested five possible sources for iron accumulation in the keratoconic corneal tissue: tears, blood plasma, breakdown of blood in perilimbal tissues, aqueous humor, and breakdown of intracellular cytochromic enzymes. However, the study conducted by Barraquer-Somers et al. [23] about different patterns and causes of iron deposition in corneal buttons, concludes that the tear film is likely the main source of iron accumulation, justifying it with the relative tear pooling mechanism on an irregular surface as responsible of the iron lines formation. Clearly, an altered iron metabolism occurs in keratoconic corneas, and this iron deposition may lead to an oxidative microenvironment and cell damage or death cell processes such as ferroptosis [24,25]. Knowing that tear iron transport among tissues is carried out by iron-binding proteins [26], the reduced tear and serum LTF levels observed in KC could contribute to iron filtration and accumulation in the corneal epithelial tissue. Similarly, other tear iron-binding proteins like serotransferrin have been shown also to be downregulated in KC patients [25], adding another contributing factor to iron deposition in corneal tissue. In this line, it could be possible to hypothesize that the oxidative microenvironment, the cell damage, and the ferroptosis induced by the iron deposition may affect the limbal cellular niche. Subsequently, it could affect the cells populations that share a common location in the palisades of Vogt at the limbal region, including the limbal stem cells (LSCs), the corneal stromal stem cells (SSCs), and the limbal stromal fibroblasts. According to this hypothesis, iron dyshomeostasis could influence the proper functioning of LSCs in the task of epithelial renewal, and the proper role of corneal stromal cells in the task of producing collagen fibrils, which collectively could induce the tissue degradation that leads to KC pathogenesis. However, further studies are needed to refine our current understanding of whether and how iron deposits in the epithelial basement membrane cause damage, by oxidative or ferroptosis processes, to the corneal epithelial cells and the surrounding ones.

The LTF involvement in other pathophysiological mechanisms of the KC disease is also uncertain. In this regard, our study focused on the immunomodulatory activity of LTF and its possible relationship with the innate immune predictive biomarkers recently discovered for KC, specifically the Toll-like receptors (TLRs). TLRs are immune innate transmembrane proteins that detect the presence of exogenous and/or endogenous agents associated with cell damage [18], triggering an inflammatory response and promoting molecular drivers which may cause tissue degradation. Overexpression of TLR2 and Toll-like receptor 4 (TLR4) in corneal and conjunctival epithelial cells was found in patients with clinical KC, subclinical KC, first-degree relatives (without abnormal clinical–topographic–aberrometric–tomographic parameters) of KC patients, and even patients with pellucid marginal degeneration (another sort of corneal ectasia) when compared to healthy control participants [17,27,28]. TLR2 & TLR4 biomarkers showed a great potential to monitor early KC changes, and demonstrated relevant roles as predictive, diagnostic, and prognostic targets for corneal ectatic disorders. Moreover, this overexpression was also observed at the blood level (in monocytes and neutrophils), showing a clear association with inflammatory serum biomarkers [16]. Both receptors acquired significant importance in the pathophysiology of the disease, however, TLR2 seems to be slightly more involved in KC. To the best of our knowledge, this is the first study that examines the relationship between TLR2 expression and LTF levels in KC patients at both systemic and ocular levels.

Regarding the immunomodulatory capacity of LTF, it has been described that it promotes the modulation of signaling molecules associated with the immune innate and adaptative response balancing [12,29]. In this linen, LTF has shown a linked activity with pathogen-associated molecular patterns involved in the TLR4 pathway. Concretely, LTF interferes with CD14-LPS interaction, resulting in the inhibition of LPS-induced TLR4 activation. This LTF interaction modulates the immunoinflammatory process mainly by preventing the release of proinflammatory cytokines and limiting the recruitment of immune cells to inflammatory sites, as well as their activation [30,31]. However, LTF is also able to promote the recruitment and maturation of immune B and T cells [29,32]. An extensive in vitro overview of the LTF interactions has demonstrated that LTF may promote or inhibit TLR4 activation, being able to moderately activate it by direct interaction and to attenuate it by CD14-LPS interaction. In addition, the NF-κB activation guided by LTF could be done by both MyD88-dependent or -independent pathways [33]. These findings suggest that LTF modulation on TLR4 pathways might be present in KC patients, but there are no studies evaluating this issue. About TLR2 modulation, there is no information about the LTF impact or about how it affects the immune cell responses. Nevertheless, after proving the association between low LTF concentrations and TLR2 overexpression at both systemic (blood samples) and local (tear fluid & ocular surface) levels in KC patients, our in vitro study with cell cultures aimed to determine the LTF’s immunomodulatory capacity on TLR2.

A fundamental hurdle in the study of the molecular-based mechanisms that lead to the development and progression of KC is the absence of a consolidated in vitro or in vivo model. In this study, we carried out an initial in vitro approach with a specifically designed cell line (HEK-Blue^TM^hTLR2) for monitoring TLR2 agonists and antagonists, in order to find out how LTF contributes to TLR2 modulation. As a result, we confirmed that the use of LTF downregulates the expression and functionality of TLR2. More specifically, in stimulated HEK-Blue^TM^hTLR2 cell cultures, TLR2 protein expression was statistically reduced using 2 mg/mL of human LTF. Moreover, the amount of SEAP and IL-8 released into the supernatant in response to the LTF immunomodulation was likewise reduced. The use of HEK-Blue^TM^hTLR2 cell line allowed us to obtain a precise, specific, and direct demonstration that LTF immunomodulation involves the NF-kB-mediated TLR2 signaling pathway, confirming the successful and functional interaction between LTF and TLR2, and providing a stronger demonstration of the TLR2 immunomodulation by the human LTF. Undoubtedly, a deeper study about LTF modulatory properties of TLR2’s immune–inflammatory response in a corneal epithelial cell line will be needed as the next step in future studies to understand in more detail the specific molecular process. This finding provides clear evidence that human LTF modulates the immune–inflammatory process mediated by the NF-κB-dependent TLR2 signaling pathway, preventing the overexpression of innate immune receptors, minimizing the release of pro-inflammatory cytokines, and limiting the enhancement of the chronic immune–inflammatory state. The ability to reduce the chronic immune–inflammatory microenvironment in KC corneas will help to minimize the cascade of molecular events (involving interleukins and metalloproteases (MMPs)) that trigger the corneal tissue degradation characteristic of the pathophysiology of the disease.

All these findings indicate a successful and functional interaction between LTF and TLR2 pathways; nevertheless, a deeper study of LTF modulatory properties on TLR2’s immune–inflammatory response in a corneal epithelial cell line is needed to understand in more detail the specific molecular process.

The use of LTF in the development of therapies to treat KC has been taken into account in recent studies. Agwa et al. [13] highlighted the safety properties provided by LTF as a natural protein that plays a vital role in many physiological processes. This work supports the idea that the structure of LTF constitutes an advantage for administration efficiency, facilitating the design of a wide range of delivery systems [13]. Recently, two types of chitosan-based nanoparticles have been proposed as novel topical ophthalmic drug delivery systems which incorporate LTF as a pharmacological alternative for KC therapy [34]. Similarly, Pastori et al. [35] proposed the use of LTF-loaded therapeutic contact lenses to exert antioxidant activity on epithelial cells to reduce oxidative stress and to provide an effective device against the KC progression. However, none of these therapeutic approaches has yet been tested in a clinical setting. The findings observed in our study could impact on biotechnology development, opening the way for going a step further with dual LTF and TLR2-blocking therapeutic approach, or as a method to better target KC treatment. The use of LTF as a therapeutic agent would help to improve not only defensive and iron uptake capacities, preventing iron accumulation, increased oxidative stress, and avoiding cellular damage; but it would also improve the immune function on the KC ocular surface.

We have identified limitations to the clinical study. All participants should be included in both blood and tear collection to correlate serum and tear levels of LTF, rather than using two separate cohorts.

In conclusion, this study evidence that serum and tear LTF levels in KC patients strongly correlate with the disease’s immune, inflammatory, and clinical status. Furthermore, the in vitro study confirmed that human LTF could indeed immunomodulate TLR2 expression in a functional manner, suggesting a potential therapeutic approach for future research.

## 4. Materials and Methods

### 4.1. Clinical Study

A total of 90 KC patients and 60 age- and gender-matched healthy controls were enrolled. Participants underwent a clinical exam with blood and tear collection. The main inclusion criterion was the KC diagnosis supported by slit-lamp, topographic, aberrometric and tomographic examination, following the KC diagnostic standards outlined in the literature [36,37,38,39]. Inclusion criteria for the healthy control participants included normal clinical parameters without biomicroscopic signs of KC, no alterations in the slit-lamp examination, no irregular astigmatisms in the tomographic evaluation that could suggest a subclinical state of the disease, and no family history of KC. Common inclusion criteria for both groups: (1) Schirmer ≥ 15 mm in 5 min; (2) conjunctival hyperemia < 2 (Nathan Efron scale) [40]; (3) at least 1 week without contact lenses and/or no instillation of eye drops. Exclusion criteria for both groups: (1) existence of active systemic or ocular inflammation, and/or current treatment with systemic or local anti-inflammatory drugs; (2) hepatic, renal, hematologic, and/or immunologic diseases, disorders of thyroid function, uncontrolled diabetes, and infections in the days preceding to the sample collection; (3) dry eye; (4) solid tumors or surgery interventions, since they may interfere with the results of the study of molecular markers. All clinical examinations were performed by the same researcher, and the molecular determinations were carried out in a laboratory blinded to clinical data.

#### 4.1.1. Blood Sample Extraction and Analysis

Blood samples were extracted by venipuncture and collected in EDTA-anticoagulated tubes. These samples were used to measure serum concentrations of LTF. For the determination of LTF serum levels, blood samples were centrifuged for 15 min at 3000× *g* and stored at −80 °C. LTF levels were measured using commercial ELISA kits following the manufacturer’s instructions (Assaypro LLC, St. Charles, MO, USA). The intra-assay and inter-assay coefficient of variation (CV) was less than 8%.

#### 4.1.2. Tear Sample Collection and Analysis

Schirmer strips (Contacare Ophthalmics Diagnostics, Guajarat, India) were used to collect the tear fluid from the temporal area of the lower eyelid. Tear collection was carried out with adequate environmental conditions, without previous administration of drugs, vital dyes, or other eye drops; gloves were used to avoid contamination of the samples. One strip was used for each participant, collecting 15 mm on the strip’s scale. Samples were frozen at −80 °C until the molecular evaluation. These samples were used to evaluate the concentrations of LTF in the tear fluid. LTF levels were measured using a commercial ELISA kit following the manufacturer’s instructions (Assaypro LLC, St. Charles, MO, USA). The intra-assay and inter-assay CV were less than 8%.

#### 4.1.3. LTF Correlation Study with Immune, Inflammatory and Clinical Variables

All participants enrolled in this clinical study had also previously participated in other studies conducted by our group. Therefore, our database (*Corneal Ocular Pathology Line (2019/623)*) collects information about their clinical status and a wide range of systemic and local immunoinflammatory variables. In this line, LTF correlation analysis was elaborated using previously collected information about: (1) TLR2 expression in blood monocytes and neutrophils, and in corneal and conjunctival epithelial cells; (2) serum inflammatory mediators (interleukin-1β (IL-1β), interleukin-6 (IL-6), matrix metalloproteinase 9 (MMP9), and tumor necrosis factor-alpha (TNF-α)); (3) clinical considerations like the presence of ocular itching and/or rubbing, and corneal topographic, aberrometric and tomographic parameters (paracentral infero-superior dioptric difference (I-S), coma, coma-like, and posterior elevation). Assessment methodology and data obtained for these collected variables have been previously published [6,7]. Briefly, TLR2 expression in blood monocytes and neutrophils (from blood samples), and TLR2 expression in corneal and conjunctival epithelial cells (from ocular surface cellular samples collected by superficial scraping with ophthalmic surgical lancets) were analyzed using FACSAria iiu flow cytometer (BD Biosciences, Franklin Lakes, NJ, USA) with FACSDiva software (BD Biosciences, Franklin Lakes, NJ, USA). TLR2 proteins were marked with fluorescein isothiocyanate (FITC) anti-TLR2-conjugated monoclonal antibodies (Immunostep, Salamanca, Spain). The mean expression of TLR2 was reported as arbitrary fluorescence units (AFU). For the determination of IL-1β, IL-6, MMP9, and TNF-α serum levels, blood samples were centrifuged for 15 min at 3000× *g* and stored at −80 °C. MMP-9 concentrations were measured by ELISA kit following the manufacturer’s instructions (GE Healthcare, UK). IL-1β, IL-6, and TNF-α levels were analyzed using an immunodiagnostic IMMULITE 1000 System (Siemens Healthcare Global, CA, USA). Finally, TOPCON CA-100 topographer-aberrometer and Orbscan IIz tomographer were used to compile the corneal quantitative parameters.

### 4.2. In Vitro Study

In vitro research was performed using the HEK-Blue^TM^ hTLR2 cell line, a cell culture specifically designed for studying the stimulation, neutralization, and immunomodulation of TLR2. HEK-Blue™ hTLR2 cell line from InvivoGen (Cat.#hkb-htlr2, InvivoGen, Toulouse, France) was obtained by co-transfection of hTLR2 and secreted embryonic alkaline phosphatase (SEAP) reporter genes into human embryonic kidney 293 (HEK293) cells. The SEAP reporter gene is placed under the control of the IFN-β minimal promoter, fused to NF-kB and AP-1-binding sites; so, when a TLR2 ligand activates NF-kB and AP-1, the production of SEAP is induced. HEK-Blue^TM^ Null1, the parental cell line of HEK-Blue^TM^ hTLR2, was used as negative control. HEK-Blue™ Null1 (Cat.#hkb-null1, InvivoGen, Toulouse, France) was obtained by transfection of SEAP reporter genes into HEK293 cells, without hTLR2-reported genes.

Both HEK-Blue^TM^ cell lines were seeded, cultured, maintained and propagated according to the manufacturer’s instructions. At 48 h before starting the experiments, cells were cultured at a density of 3 × 10^5^, 1 × 10^5^, or 4 × 10^4^ cells/well in respectively 12-, 24-, or 96-well plates (depending on the experiment). As a rule, each experiment of the in vitro study comprised at least three replicates for each measurement.

#### 4.2.1. Experimental Design

To achieve TLR2 overexpression, HEK-Blue^TM^ cell cultures were incubated for 18 h with Pam2CSK4 (10 ng/mL, InvivoGen, Toulouse, France) or Pam3CSK4 (100 ng/mL, InvivoGen, Toulouse, France), both synthetic lipopeptides TLR2 agonists. An IgA monoclonal antibody to human TLR2 (Anti-hTLR2-IgA, 10 µg/mL, InvivoGen, Toulouse, France) was used to neutralize TLR2 expression. The neutralizing protocol was performed following the manufacturer’s instructions; briefly, HEK-Blue™ cells were pre-incubated for 1 h with Anti-hTLR2-IgA and then stimulated with Pam2CSK4 or Pam3CSK4 for 18 h. To assess the immunomodulatory activity of LTF, HEK-Blue^TM^ cell cultures were pre-incubated with different concentrations (0.01–2 mg/mL) of human LTF (L1294, Sigma, St. Louise, MO, USA), following the same method carried out for neutralizing. 2 mg/mL of human LTF was the dose that achieved the best neutralization capacity, shown in Appendix A, and so it was used for the in vitro TLR2 immunomodulation study. TLR2 overexpression, neutralization, and immunomodulation were first determined by measuring SEAP production with QUANTI-Blue^TM^ assay. Flow cytometry, immunohistochemistry, WB, and ELISA experiments were then performed to validate the expression and functionality of TLR2.

#### 4.2.2. QUANTI-Blue^TM^

QUANTI-Blue^TM^ (rep-qbs, InvivoGen, Toulouse, France) is a colorimetric enzyme assay developed to detect SEAP in cell supernatant. The supernatant of unstimulated, stimulated, neutralized, and immunomodulated HEK-Blue^TM^ cell cultures were collected for SEAP quantification by QUANTI-Blue^TM^ assay. The measurements were carried out in 96-well plates, and SEAP levels were quantified using a spectrophotometer (Synergy 2, BioTek, USA) at a 655 nm wavelength, according to the provided instructions.

#### 4.2.3. Flow Cytometry

The unstimulated, stimulated, neutralized, and immunomodulated HEK-Blue^TM^ cell cultures were trypsinized (trypsin-EDTA, Gibco, Gaithersburg, MD, USA) and removed from the 24-well plates in which they were seeded. After 5 min centrifugation at 200× *g*, the cell pellet was resuspended in D-PBS (ATCC, Virginia, USA) and incubated for 20 min with FITC anti-TLR2-conjugated monoclonal antibodies (5 µL, Immunostep, Salamanca, Spain). TLR2 expression was evaluated using FACSAria iiu flow cytometer (BD Biosciences, Franklin Lakes, NJ, USA) with FACSDiva software (BD Biosciences, Franklin Lakes, NJ, USA). To avoid that the background from the antibody may interfere with the study, samples were washed with D-PBS and centrifuged before being analyzed. Unlabeled and thus non-fluorescent cells, as well as HEK-Blue^TM^ Null1 cells, were used as both negative controls. TLR2 expression was reported as AFU.

#### 4.2.4. Immunohistochemistry

Immunohistochemistry was automatically performed using an AutostainerLink 48 immunostainer (Dako-Agilent, Santa Clara, CA, USA). Briefly, the slides were incubated at room temperature (RT) in: (1) mouse monoclonal antibody to TLR2 (ab9100) (Abcam, Cambridge, UK) at 1:100 for 30 min; (2) EnVision^®^+ Dual Link System-HRP (dextran polymer conjugated with horseradish peroxidase and affinity-isolated goat anti-mouse and goat anti-rabbit immunoglobulins) (Dako-Agilent, K4065) for 20 min; (3) DAB+ substrate-chromogen solution (1 mL of substrate buffer solution containing hydrogen peroxide and 20 µL of 3,3′-diaminobenzidine tetrahydrochloride chromogen solution) for 10 min; and (4) EnVision FLEX hematoxylin for 15 min. The intensity of each TLR2 staining was quantified by ImageJ (Rasband, WS, USA) by measuring the inverted DAB signal and by calculating the average with at least thirteen cells for each image.

#### 4.2.5. Western Blot

The unstimulated, stimulated, neutralized, and immunomodulated HEK-Blue^TM^ cell cultures were trypsinized and removed from the 12-well plates in which they were seeded. After centrifugation, the cell pellets were resuspended in RIPA lysis buffer (Thermo Scientific, Waltham, MA, USA) with protease inhibitor (Roche, Suiza) and shacked for 15 min on ice. The lysates were collected and centrifuged at 21,000× *g* for 30 min at 4 °C, and the supernatants were collected and stored at −80 °C. To prepare samples for WB, the total concentration of proteins was quantified using a micro BCA protein assay kit (Thermo Scientific, Waltham, MA, USA) according to the manufacturer’s instructions. Each sample was aliquoted with 15 µg of total proteins, combined with 4× loading buffer, and denatured in a 95 °C metal bath for 10 min. Samples were run on SDS-PAGE 10% protein gel using a fixed voltage of 140 V. Proteins were electro-transferred to a PVDF membrane (Amersham^TM^ Hybond 0.45, USA) using a Trans-Blot semi-dry system (Bio-Rad, Hercules, CA, USA) with a limited voltage of 25 V and 180 mA for 1 h 40 min. Post-blot membranes were blocked for 45 min with 3% bovine serum albumin (BSA, Sigma, St. Louise, MO, USA) (3% BSA diluted in Tris chloride Buffered Saline with Tween^®^ 20 (TBST)), and then incubated overnight at RT in agitation with 1:500 rabbit polyclonal Anti-TLR2 target protein (A11225, ABclonal, Woburn, MA, USA) and with 1:2000 rabbit polyclonal Anti-β-actin control protein (Abcam, UK), both diluted in 3% BSA-TBST. TLR2 has a molecular weight of 89 kDa, while β-actin weighs 42 kDa. After overnight incubation, membranes were washed three times in TBST and incubated again for 1 h at RT in agitation with 1:5000 HRP-conjugated goat anti-rabbit IgG secondary antibody (Dako, Denmark) diluted in 3% BSA-TBST. Finally, membranes were washed in TBST, and revealed with Pierce™ ECL Western Blotting Substrate (Thermo Fisher, Lexington, MA, USA). ChemiDoc^TM^ MP imaging system (Bio-Rad, Hercules, CA, USA) was used for band detection. Results were analyzed by ImageJ (Rasband, WS, USA) by measuring the mean grey value of protein bands delimited in ROIs. Relative expression of TLR2 to β-actin was calculated for each sample, and each group sample average was normalized to the control.

#### 4.2.6. ELISA

The supernatant of unstimulated, stimulated, neutralized, and immunomodulated HEK-Blue^TM^ cell cultures seeded in 24-well plates, was collected for IL-8 quantification. IL-8 concentrations were quantified using an ELISA kit (EK0413, Boster, Pleasanton, CA, USA) according to the manufacturer’s instructions.

### 4.3. Statistical Analysis

SPSS 20.0 for Windows (IBM, NY, USA) was used to conduct the statistical analysis. A Kolmogorov–Smirnov test was used to verify the normality of a quantitative variable. Results were expressed as mean (±standard deviation (SD)) for continuous quantitative variables with normal distribution, and as percentages for categorical variables. Bivariate comparisons were made with Student’s *t* test (normal continuous variables), and with the χ2 test (categorical variables). Graphic representations of the comparisons between normal continuous variables were made using error bars. An ANOVA test was used to make comparisons among more than two study groups following a DMS post hoc test. Bivariate correlations for normal distribution were analyzed using Pearson’s coefficients and represented by scatter plots. A *p* value less than 0.05 was considered significant in all tests. The statistical EPIDAT 3.1 software was used for calculating the sample size of the clinical study. This determination was based on preliminary published studies of TLR2 levels [6,7]; accepting a confidence level of 95% (α = 0.05) and 75% power (β = 0.25) of at least 20 control eyes and 40 eyes with KC would be necessary.

## Figures and Tables

**Figure 1 ijms-23-12350-f001:**
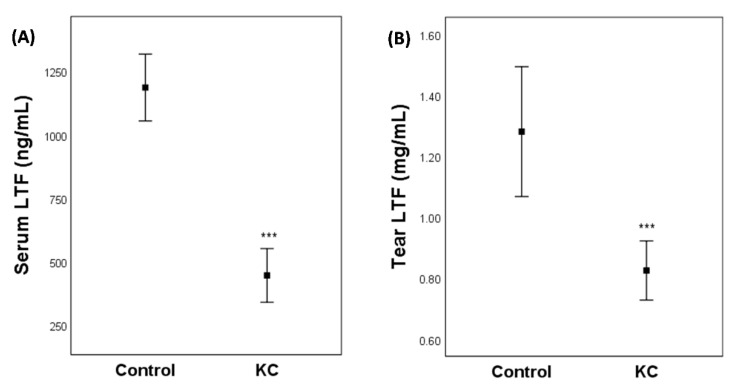
Serum (**A**) and tear (**B**) LTF concentrations in control subjects and KC patients. Statistical differences with regard to controls: *** *p* < 0.0001. Sample size: (**A**) controls = 20 eyes, KC = 40 eyes; (**B**) controls = 40 eyes, KC = 50 eyes. Abbreviations: KC, keratoconus; LTF, lactoferrin.

**Figure 2 ijms-23-12350-f002:**
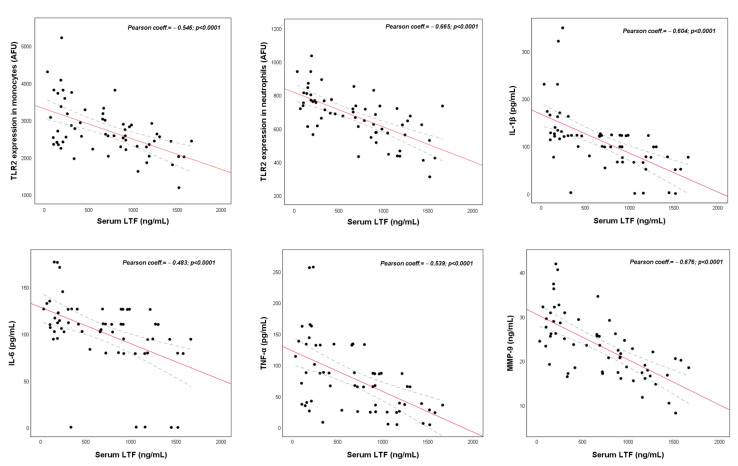
Scatter plots between serum LTF levels and immune–inflammatory biomarkers studied in blood samples showed a strong negative correlation for all of them. Abbreviations: AFU, arbitrary fluorescence units; IL, interleukin; LTF, lactoferrin; MMP-9, matrix metalloproteinase 9; TLR2, Toll-like receptor 2; TNF-α, tumor necrosis factor-alpha. Legend: the black dots represent the study cases, the grey dotted lines represent the mean confidence intervals, and the red line represents the linear fit line.

**Figure 3 ijms-23-12350-f003:**
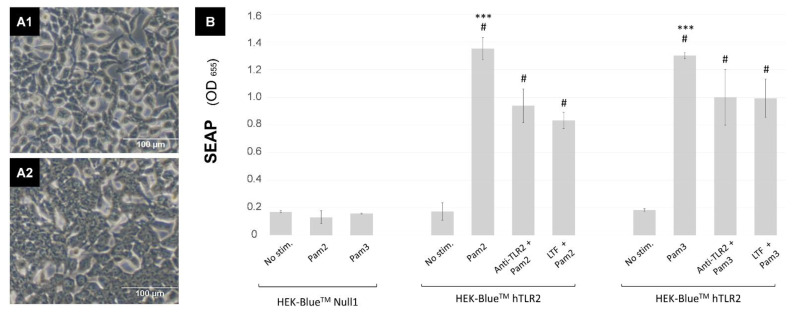
(**A**) Brightfield microscopy image of HEK-Blue^TM^ cells at a 20x magnification; (**A1**) transfected with hTLR2 (HEK-Blue^TM^hTLR2) and (**A2**) without hTLR2 (HEK-Blue^TM^Null1). (**B**) SEAP concentrations measured by QUANTI-Blue^TM^ assay in the supernatant of unstimulated, stimulated, neutralized, and immunomodulated HEK-Blue^TM^ cell cultures; statistical differences regarding to: # *p* < 0.0001 HEK-Blue^TM^Null1 & Unstimulated HEK-Blue^TM^hTLR2, *** *p* < 0.005 LTF & Anti-hTLR2-IgA. Abbreviations: Anti-TLR2, Anti-hTLR2-IgA; LTF, lactoferrin; No stim., unstimulated; OD, optical density; Pam2, Pam2CSK4; Pam3, Pam3CSK4; SEAP, secreted embryonic alkaline phosphatase; TLR2, Toll-like receptor 2.

**Figure 4 ijms-23-12350-f004:**
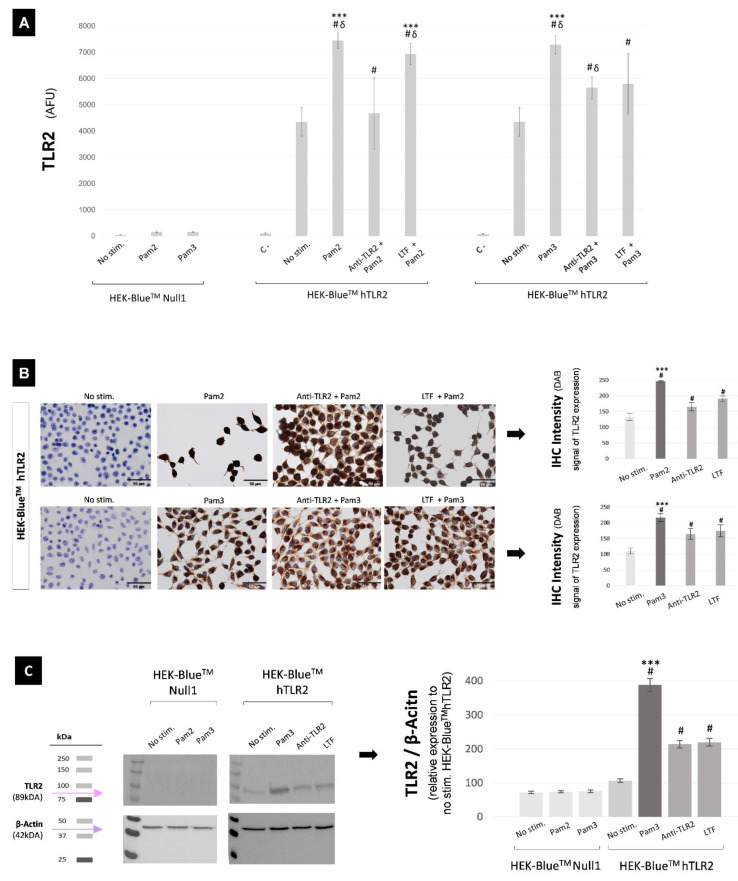
TLR2 protein expression in unstimulated, stimulated, neutralized, and immunomodulated HEK-Blue^TM^ cell cultures, measured by flow cytometry (**A**), immunohistochemistry (**B**), and WB (**C**). (**A**) Statistical differences with regard to: # *p* < 0.0001 HEK-Blue^TM^Null1 and unlabeled cells (negative control), δ *p* < 0.001 unstimulated cells, *** *p* < 0.05 all groups. (**B**) Microscopic images of HEK-Blue^TM^hTLR2 cells at a 40× magnification, and quantification of the intensity of staining; statistical differences with regard to: # *p* < 0.0001 unstimulated HEK-Blue^TM^hTLR2, *** *p* < 0.0001 LTF and Anti-hTLR2-IgA. Legend: scale bars represent 50 μm. (**C**) WB revealed membrane and quantitative results analysis; statistical differences with regard to: # *p* < 0.0001 HEK-Blue^TM^Null1 and unstimulated HEK-Blue^TM^hTLR2, *** *p* < 0.0001 LTF and Anti-hTLR2-IgA. The uncropped blots are shown in Appendix A. Abbreviations: Anti-TLR2, Anti-hTLR2-IgA; C-, negative control (unlabeled cells in flow cytometry); IHC, immunohistochemistry; LTF, lactoferrin; No stim., unstimulated; Pam2, Pam2CSK4; Pam3, Pam3CSK4; TLR2, Toll-like receptor 2; WB, western blot.

**Figure 5 ijms-23-12350-f005:**
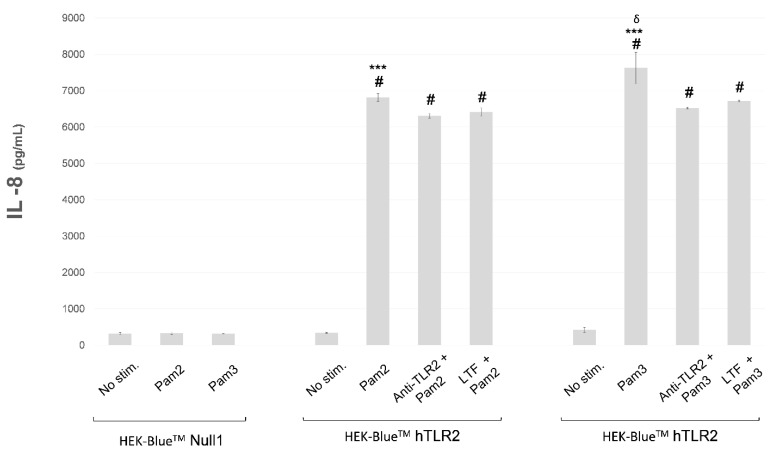
IL-8 concentrations were measured by ELISA in the supernatant of unstimulated, stimulated, neutralized, and immunomodulated HEK-Blue^TM^ cell cultures. Statistical differences with regard to: # *p* < 0.0001 HEK-Blue^TM^Null1 and unstimulated HEK-Blue^TM^hTLR2, *** *p* < 0.0001 Anti-TLR2 and LTF, δ *p* < 0.0001 Pam2CSK4 HEK-Blue^TM^hTLR2. Abbreviations: Anti-TLR2, Anti-hTLR2-IgA; IL-8, interleukin 8; LTF, lactoferrin; No stim., unstimulated; Pam2, Pam2CSK4; Pam3, Pam3CSK4; TLR2, Toll-like receptor 2.

**Table 1 ijms-23-12350-t001:** Correlations (Pearson’s coefficients) between tear LTF concentrations and ocular markers (TLR2 expression in corneal and conjunctival epithelial cells, and other corneal topographic, aberrometric, and tomographic parameters).

Tear LTF/Ocular Markers	Pearson’s Coefficient, *p* Value
Tear LTF/TLR2 cornea	r = −0.289, **=0.007**
Tear LTF/TLR2 conjunctiva	r = −0.266, **=0.01**
Tear LTF/I-S asymmetry	r = −0.282, **=0.008**
Tear LTF/Coma	r = −0.314, **=0.003**
Tear LTF/Coma-like	r = −0.330, **=0.002**
Tear LTF/Post. elevation	r = −0.227, **=0.04**

Abbreviations: LTF, lactoferrin; TLR2, Toll-like receptor 2; I-S, paracentral infero-superior dioptric difference.

## Data Availability

Not applicable.

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
