# Peer review of "Immunomodulatory Effect of Human Lactoferrin on Toll-like Receptors 2 Expression as Therapeutic Approach for Keratoconus"

_ijms, 2022, doi:10.3390/ijms232012350_

Round 1

Reviewer 1 Report

The data presented in this interesting paper are sound and the conclusions valid. As for Discussion, Fleisher rings are caused by iron deposits in basal cells in proximity to the limbus that reside limbal stem cell (LiSC) important for maintenance of corneal stromal cells that produce collagen fibrils. It may be worthwhile to discuss whether iron deposits may affect LiSC function thus contributing to keratoconus pathogenesis.

Author Response

We appreciate the reviewer’s suggestion. Certainly, it has been reported that the Fleisher ring, caused by iron deposits in the epithelial basement membrane, occurs at the base of the cone in keratoconic corneas being often located in proximity to the limbus; consequently, we agree with the hypothesis proposed by Reviewer 1. In this regard, ferroptosis and iron toxicity could be affecting the limbal stem cells (LSCs) and the niche of cell populations that share the same location in the Palisades of Vogt at the limbal region.

LSCs represent a quiescent cell population with proliferative capacity that receives molecular signals from growth factors, cytokines and other soluble molecules, and that respond to the surrounding physical structures via mechanotransduction, interaction with the extracellular matrix, and interactions with several cell types [1]. LSC represents a relatively well-studied entity with proven clinical relevance in corneal restoration and renewal of corneal epithelium [2]. Moreover, the LSCs share location (the Palisades of Vogt at the limbal region) with the corneal stromal stem cells (SSCs) and with the limbal stromal fibroblasts, that contribute to the production of stromal cells and the secretion of collagens and proteoglycans essential for the maintenance of the corneal structure [1]. In this line, the iron dyshomeostasis could affect the proper functioning of LSCs in their task of epithelial renewal, and the proper role of corneal stromal cells in the task of producing stromal collagen fibrils; which could contribute to the tissue degradation that leads to the Keratoconus pathogenesis.

After carrying out a deep search on these aspects, in order to look for a justification or explanation for this hypothesis, we have not found any published study. Therefore, we really appreciate the comment of Reviewer 1, which not only has contributed to improving the quality of our submission, but also amplifying the scope for new research projects that help understand whether and how the damage caused by the iron deposition on the epithelial basement membrane affects to the surrounding cells.

References:

[1] Robertson SYT, Roberts JS, Deng SX. Regulation of Limbal Epithelial Stem Cells: Importance of the Niche. Int J Mol Sci. 2021 Nov 5;22(21):11975.

[2] Gonzalez G, Sasamoto Y, Ksander BR, Frank MH, Frank NY. Limbal stem cells: identity, developmental origin, and therapeutic potential. Wiley Interdiscip Rev Dev Biol. 2018 Mar;7(2):10.1002/wdev.303.

As a result, the following paragraph was added to the Discussion of the manuscript:

  • Line 299: It could be possible to hypothesize that the oxidative microenvironment, the cell damage, and the ferroptosis induced by the iron deposition may affect the limbal cellular niche. Subsequently, it could affect the cells populations that share a common location in the Palisades of Vogt at the limbal region, including the limbal stem cells (LSCs), the corneal stromal stem cells (SSCs), and the limbal stromal fibroblasts. According to this hypothesis, iron dyshomeostasis could influence the proper functioning of LSCs in the task of epithelial renewal, and the proper role of corneal stromal cells in the task of producing collagen fibrils; which collectively could induce the tissue degradation that leads to KC pathogenesis. However, further studies are needed to refine our current understanding of whether and how iron deposits in the epithelial basement membrane cause damage, by oxidative or ferroptosis processes, to the corneal epithelial cells and the surrounding ones.

Reviewer 2 Report

The current manuscript entitled "Immunomodulatory Effect of Human Lactoferrin on Toll-Like Receptors 2 Expression as Therapeutic Approach for Keratoconus", the authors show the dysregulation of LTF and TLR2 in the ocular surface of KC patients contributes to KC severity by maintaining a detrimental chronic immune-inflammatory state.

The manuscript is well written and minor English language edition is needed.

Author Response

We thank the Reviewer 2 for his/her time in reviewing our manuscript.

As Reviewer 2 has suggested, a native English speaker has reviewed our article for grammar, spelling, punctuation, and writing to improve readability.

The English editions were marked in the manuscript using the “Track Changes” function of Microsoft Word.

Reviewer 3 Report

This is a very well conducted work on the grounds of the pathogenesis of keratoconus.

The article is well written, supported by a very comprehensive and well described methodology.

There are only two points to comment on:

* Why did you select HEK-BlueTMhTLR2 cell 337 cultures, instead of cell cultures from the cornea? Please comment in Material and Methods and/or in Discussion..

* Understanding that lactoferrin alters de immune responses, in which way can they provoke the ectatic changes of the cornea? This would be interesting to include in the Discussion.

Author Response

We thank the Reviewer 3 for his/her positive comments on our manuscript.

Regarding the first point (Why did you select HEK-BlueTMhTLR2 cell cultures, instead of cell cultures from the cornea?): In this study, we have chosen the HEK-BlueTMhTLR2 cell cultures because it is a cell line specifically designed for monitoring TLR2 agonists and antagonists. The use of this cell line as the first approach to evaluate the immunomodulatory effect of LTF on TLR2 expression was done in order to obtain a more precise, specific, and direct demonstration that the LTF immunomodulation was mediated through the NF-kB-activated TLR2 signaling pathway. Undoubtedly, the use of corneal epithelial cell cultures will be necessarily the next step to develop (in fact, we have our scope focused on that for our future research). The HEK-BlueTM cell line has helped to confirm, in a conclusive way, that LTF influences the desired signaling pathway, providing a stronger demonstration of TLR2 immunomodulation.

As Reviewer 3 suggests, the following information was added to the Discussion of the manuscript:

  • Line 353: In this study, we carried out an initial in vitro approach with a specifically designed cell line (HEK-BlueTMhTLR2) for monitoring TLR2 agonists and antagonists, in order to find out how LTF contributes to TLR2 modulation.
  • Line 361: The use of HEK-BlueTMhTLR2 cell line allowed us to obtain a precise, specific, and direct demonstration that LTF immunomodulation involves the NF-kB-mediated TLR2 signaling pathway; confirming the successful and functional interaction between LTF and TLR2, and providing a stronger demonstration of the TLR2 immunomodulation by the human LTF. Undoubtedly, a deeper study about LTF modulatory properties on TLR2´s immune-inflammatory response in a corneal epithelial cell line will be needed as the next step in future studies to understand in more detail the specific molecular process.

Regarding the second point (Understanding that lactoferrin alters de immune responses, in which way can they provoke the ectatic changes of the cornea?): We have demonstrated that LTF modulates the immune response. Hence, we hypothesize that the way in which LTF causes changes in the cornea may be through the ability to reduce the chronic immune-inflammatory microenvironment characteristic of KC. Specifically, human LTF modulates the immune-inflammatory process mediated by the NF-κB-dependent TLR2 signaling pathway; preventing the overexpression of innate immune receptors, minimizing the release of pro-inflammatory cytokines, and limiting the potentiation of the chronic immune-inflammatory state. In this regard, LTF helps to minimize the cascade of molecular events (involving interleukins and metalloproteinases (MMPs)) that trigger the corneal tissue degradation characteristic of the pathophysiology of the disease.

As Reviewer 3 suggest, the following information was added to the Discussion of the manuscript:

  • Line 368: This finding provides clear evidence that human LTF modulates the immune-inflammatory process mediated by the NF-κB-dependent TLR2 signaling pathway; preventing the overexpression of innate immune receptors, minimizing the release of proinflammatory cytokines, and limiting the enhancement of the chronic immune-inflammatory state. The ability to reduce the chronic immune-inflammatory microenvironment in KC corneas will help to minimize the cascade of molecular events (involving interleukins and metalloproteases (MMPs)) that trigger the corneal tissue degradation characteristic of the pathophysiology of the disease.